# Improved Behavioral Box and Sensing Techniques for Analysis of Tactile Discrimination Tasks in Rodents

**DOI:** 10.3390/s23010288

**Published:** 2022-12-27

**Authors:** José Wanderson Oliveira Silva, Renan Araújo Lima, Edgard Morya, Fabricio Lima Brasil, Luiz Marcos Garcia Gonçalves

**Affiliations:** 1Graduate Program in Computer and Electrical Engineering, Universidade Federal do Rio Grande do Norte, Natal 59078-970, Brazil; 2Instituto Santos Dumont, Macaíba 59280-000, Brazil

**Keywords:** somatosensory stimuli, rodents whiskery behavior, behavioral box

## Abstract

Neuroscience behavioral studies stand out among the research works in this area. In these studies, normally, rodents are put inside closed platforms known as behavioral boxes in order to perform tasks and have their behaviors observed by way of sensors and annotations by hand. In this work, we aim to improve this research process by developing new techniques based upon the full automation of the behavioral box processes for more easily acquiring data. We come up with a new structural design using cutting-edge technology, with enhanced spaces and better materials. We use components that can be easily purchased (or built) and developed new techniques for control and data acquisition. Our new platform allows for more precise control of the opening of the discrimination bars, which was not satisfactorily done with previous platforms. This makes possible the design of more complex decision-making experiments using camera and sensor systems, allowing a better assessment of rodent performance in the discrimination task. All the necessary materials and development documents are made available in a collaborative multi-user platform allowing work replication. With this, the present study provides a low-cost tool with ease of development and construction that can be used by laboratories that work with this type of research.

## 1. Introduction

Rodents are known to be one of the most used subjects in neuroscience experiments nowadays due to a series of characteristics, such as reduced size, short reproductive cycle, numerous offspring, precocity, easy handling, varied nutrition, and adaptation to captivity [1]. Among the various research types performed with rodents, behavioral studies on operant conditioning stand out. They usually consist of annotating and analyzing a learning process in which the animal’s behavior is modified by a reward or punishment. Nonetheless, rodents are also excellent for the study of the tactile sensory system, as these animals use their vibrissae to locate and discriminate objects. The tactile sensory system is of fundamental importance for animals that have nocturnal habits and depend mainly on their vibrissae to locate and discriminate objects. The prefrontal cortex, more specifically the anterior cingulate, is associated, among other functions, with error monitoring, attention, and strategy change in dual-choice tasks, influencing decision-making processes. The decision-making process is a complex cognitive function that depends on the integration of diverse sensory stimuli and previous experiences to anticipate and improve an outcome.

These studies are carried out, in general, using a training apparatus known as an operant conditioning box (OCB). There are many types of OCBs, and the functioning of each type aims to study some desired characteristics of the behavior of the animals. For example, using T and Y mazes [2] normally applies to studying the decision-making process. The elevated-plus maze aims to study anxiety characteristics, while the open field ones aim to study both spatial and anxiety characteristics [3]. There are two main issues with the current platforms, which are their cost and their availability. The availability is a natural consequence of the cost, most of the time they are not easily available as commercial platforms because there are not many consumers. Yet, some of them are specifically designed and built by proprietary houses, without technical sheets available, and there is a need for a specialized professional from the same industry/house, in order to repair them in case of problems.

One of these houses is Plexon [4], which is a pioneer and leading innovator of high-performance custom solutions for data acquisition, behavioral research, and advanced analytics designed specifically for neuroscience research. Its equipment and solutions lay the foundation for work being done around the world in areas such as basic science, brain–machine interfaces (BMI), neurodegenerative diseases, addictive behaviors, and neuroprosthetics. Plexon’s customers represent some of the most advanced national and international academic facilities, research hospitals, pharmaceutical companies, and government laboratories. Based on research using Plexon products, neuroscience labs have published thousands of articles, many of which are found in the pages of neuroscience’s most prestigious peer-reviewed journals such as Science, Nature, and Neuron [4].

Nonetheless, we noticed that by using these commercially available apparatuses, most of the time the researcher remains a slave of that technology and house client, paying unnecessary money for services that a simple lab technician with basic experience in electrical and electronics and/or some experience in computing platforms could perform. That brings up another problem, due to their price, they are not easy to be upgraded or substituted with modern equipment, and most of the time are outdated in the labs. With this, hand-done tasks are a time-consuming part of the experiments, even with manual annotation of the variables in some of the tasks.

In order to overcome or diminish such problems, in this work, we aim to study ways for improving the behavioral box including its design and construction from scratch, and eventually the techniques used in this research area. We propose a new model, with a new system for automating most tasks in this kind of platform and for taking data from it, coming up with an alternative way to perform tactile discrimination tasks. Our main contribution is the new design, with a new structure that has better-elaborated spaces and uses cutting-edge materials and technology, with components that can be purchased on the market or easily developed.

Our proposal improves the study of behavioral models related to the sensory-motor system in a more refined way. The new platform allows more precise control of the opening of the discrimination bars, which is currently not done satisfactorily by current platforms. Further, it allows the design of more complex decision-making experiments using both the camera and sensors’ system, allowing a better assessment of rodent performance in the discrimination task. Therefore, the present study provides a step further in this area, with a low-cost tool with ease of construction and operation that can be used by laboratories that work with this type of research. We have available all the necessary materials and development documents. All developed code is on the GitHub multi-user collaborative platform (github.com/renanaraujoeng/behavior_box, accessed on 21 December 2022) and the documents, printer project, and pictures can be retrieved from our site (natalnet.br/sibgrapi2022/behavioral-box, accessed on 21 December 2022).

Thus, in this work, we range from the theory of conditioning behavior to the description of the behavioral apparatus and its autonomous functioning process (automation). We analyze the most relevant theory to develop an apparatus that can adapt to the needs of the study of learning, on the sensorimotor system, including modularity, portability, and capability of performing multiple experiments, among other topics. The design stages of the components that make up the box and their functioning are clearly specified. In the following sections, we depict the theory necessary for understanding the new model of our proposal and the methodology associated with it. We further provide details on the construction and experiments with results that demonstrate its correct operation, followed by concluding remarks.

## 2. Theory and Methodology

The study of the tactile sensory system of rodents uses behavioral boxes [5], which are a specific type of OCB in which the animals have to discriminate the width of an opening using only their vibrissae, normally receiving a reward when it touches some places [6]. When dealing with operant conditioning, the used tasks can be classified as simple or conditional discrimination, as described next. We also need to understand some basics of automation, including instrumentation models, and automated and supervisory systems. At the end of this section, the description of behavioral device usage in tactile discrimination tasks is provided, thus completing the theory and methodology.

### 2.1. Operant Conditioning

Operant conditioning can be understood as a technique devised in computational neuroscience that can be applied to understanding the learning of new behaviors. This learning takes place through the use of stimuli (auditory, visual, tactile, or gustatory) and through environmental consequences (reinforcements and punishments). Since its creation, it has been used in several areas of knowledge, besides neuroscience. Its use has opened doors to a wide range of new knowledge through tasks that emulate behaviors.

Simple discrimination is one of the developed methodologies used in operant conditioning. The materialization of its contingency and the proof of the learning method were developed from the work of Skinner [5]. Skinner was a behavioral psychologist who developed an environment called the Skinner box, composed of a lever, a feeder, and a grid that generates an electric shock. In this environment, two types of contingencies could be observed. If when the lever is presented, the individual performs the behavior of pressing it, then he will receive the reward (reinforcement). If when the lever is presented, the individual performs the behavior of going to the back of the box, then he will receive a small electric shock (punishment).

Experimental paradigms measuring response time often assess how the information process that includes a sequence of perception, decision-making, and action is organized [7]. For the development of behavioral tasks that contemplate the evaluation of these processes, different experimental models have been developed, such as configuration, single-rule, and multiple-rule models [8]. These were developed through the establishment of conditional discrimination. Unlike respondent conditioning, in which a stimulus elicits a response, operant conditioning stimuli have the function of creating a context for a new response to occur.

### 2.2. Device Automation

The main characteristic of automation is to have tasks performed by machinery, which at its beginning had the purpose of increasing production efficiency [9]. This goal has changed a little nowadays, and more specifically in the current work, the idea is to use automation to accelerate and make it easier the study of behavioral conditioning with rodents. Currently, any automated system is composed of two main parts [9]:(1)**Operational Part**—It is the part of the system that acts directly on the process and is a set of elements that make the machine move and perform the desired operation. These elements that form the operational part are the drive and pre-drive devices such as motors, cylinders, compressors, valves, pistons, and also detection devices such as inductive sensors, capacitive sensors, vision sensors, and ultrasonic sensors, among others [9].(2)**Control part**—In the control part, we have the programmable tool of the system that is usually implemented with the help of a PLC (programmable logic controller). In the past, this logic was done with electromagnetic relays, timers, electronic boards, and logic modules. Currently, with the increase in the volume of data and electronic components, the most common is the use of PLCs and industrial computers to control machines and processes [9].

Automatizing a behavioral box requires some kind of instrumentation, which can be understood as any device (instrument) or set of devices used for the purpose of measuring, indicating, recording, or controlling the variables of a process [10]. There can have specific instruments for measuring, indicating, recording, and controlling each process variable. Thus, the automation process can be understood as the set of operations necessary to obtain a product or control a machine operation. Industrial processes are similar to those of the human body, but we start with the decision of the desired degree of automation. This is because, in the human body, several processes are automatic regardless of the decision, such as breathing, blood circulation, the processes that involve digestion, and other biological needs.

In the case of industrial systems, in general, automation requires the use of a supervisory system, which allows the monitoring and tracking of information. Data are collected through data acquisition equipment and then manipulated, analyzed, stored, and subsequently presented to the user all in real time. Our proposed box system uses a similar approach to monitor and supervise the variables and devices of its control system that are connected through specific controllers (drivers). The information should be continuously updated and stored in a database that can be local or remote. General supervisory systems were created to reduce the size of the panels and improve the human/machine interface; they are based on computers running specific industrial process supervision software. In our work, a much simpler system is used to perform the same three basic activities of supervision, operation, and control. The behavioral box device consists of a platform that is constructed with several components such as motors, sensors, and microcontrollers. These elements are fundamental for us to carry out the automation of the given platform operation. Interaction with it is carried out through inputs and outputs, basically. Aiming at the need to monitor training and optimize all the functional stages of the device, we developed two systems, one for control (calibration) and another for the supervision of the behavioral box itself, as it will be described further.

### 2.3. Using the OCB for Tactile Discrimination Tasks

A previous OCB model for tactile discrimination tasks was conceived and developed by Krupa and colleagues [6]. It is basically a Plexon^TM^ based platform that presents a mechanism that allows the animals to discriminate the opening between two bars using only their vibrissae. The animal has to associate the wide or narrow opening with the position (right or left side of the box) where it receives a drop of water as a reward. That OCB has been further used in tactile discrimination experiments [11], where brain–brain interfaces are used to help rodents to collaborate with each other. When a second rodent is unable to choose the correct lever, the first one notices (not getting a second reward) and produces a round of task-related neuron firing that makes the other more likely to choose the correct lever. Both are rewarded when actions are performed by the “decoding rodent” in conformation to the input signals and when the signals are transmitted by the “coding rodent”, which results in the desired action.

However, the development of this box is almost 20 years old, and electronic components and actuators that are already technologically outdated were used for its construction. Nonetheless, the work developed by Krupa and collaborators [6] allowed a great advance in the understanding of the rodent somatosensory system. Still, there are other works that show that the nose region of the rat primary somatosensory cortex (SI) integrates information from contralateral and ipsilateral whisker pads. In order to overcome the technological difficulties, we envision the possibility of developing a low-cost portable device that uses components present in the market and that allows the rodent to train tactile discrimination with greater efficiency and more resources. Nowadays, the operation of the box is a complex procedure, involving controller boards connected to a dedicated PC, an air compressor for actuators driving, and a PCB with numerous components affixed to a frame of more than 1 m^2^. In addition, this outdated (previous) platform used numerous components that are very rare to be found nowadays. Besides these problems, one of its advantages is the concept of open hardware that was adopted, which means that anyone can assemble, modify, improve, and customize it, starting from the same basic hardware.

Nonetheless, the development of a new behavioral box model with cutting-edge technology and new methods allows for greater flexibility and greater diversification in training, in addition to the classic performed by Krupa, such as forced and unilateral training. It makes it possible to design more complex experiments within the same model, such as:Small variations in the opening of the bars and how this variation behaves in the rat’s brain firing, as well as facilitating the programming of the OCC.Operant conditioning tests:Taste aversion (replacing the sweet reward with something the rat does not like or something neutral);Preference for sucrose (sucrose in a nose poke and water in another one).One can use working memory tests or add shock by electrical pulses, in order to use aversive memory training.Reaction of the sensorimotor system with variations in the opening of the bars alternately, where only one bar is extended and the other contracted, and how this variation behaves in the brain firings of the rat, observing that only stimuli could be captured by only one side of the vibrissae in the CCO.As it is a modular platform, we can make variations in the locations of the nose pokes with their specific purposes, and we can obtain new experimental models for different fields of study.

## 3. Conceptualizing and Developing a New CCO Model

Hence, in the current work, we used the above-mentioned OCB as the baseline and upgraded the hardware to the Arduino Mega microcontroller, which is a board based on the Atmel SAM3X8E ARM Cortex-M3 CPU with a 32-bit ARM core microcontroller [12]. It is connected to the host computer and programmed via its integrated development environment (IDE) using the C/C++ language, without the need for extra equipment, just using a USB cable. This microcontroller has independent (concurrent) functions, e.g., one can use it to control the opening and closing of gates inside the box at the same time that it is being used as a measuring (sensor input) device. Thus, in the project developed here, the Arduino is the hardware responsible for automating and controlling the operating processes of the behavioral box, being responsible in a specific way for controlling the opening and closing of the sliding door, controlling the rewarding release of the nose pokes, and for reading sensors input.

All aspects of the training are fully automated, controlled by Arduino and Python, and require no intervention from the experimenter. The operation is controlled by the supervisory software developed in Python with the mechatronics part done by the Arduino IDE. The behavior of each rodent can be monitored via infrared-sensitive video cameras placed above both the discrimination chamber and the main chamber. This eliminates any need to occlude the rodent’s view. In the same way, the construction of a device composed of several simple replacement mechanisms is relevant in its handling and operation. All of the issues involved with the improved platform, including its proposed new design, construction, and functioning will be treated next in this section.

### 3.1. Behavioral Box Schematic Design

The new open-hardware automated OCB model to be used in tactile discrimination experiments with rodents needs to have portability, functional variety, efficient performance, and mainly low cost. All actuators are to be built with servo motors and stepper motors, which allows for easy replacement and portability. Therefore, the new behavioral box has been designed in such a way as to eliminate the possibility that the rats could use other sensory cues (e.g., auditory, visual, or olfactory) to solve the task or, more importantly, that the rats could use their whiskers and nose to perform the tasks. With this, tactile discrimination tasks and any other training can be addressed. In addition, the behavioral apparatus is designed for the rats to approach and sample the variable-width opening with their large facial whiskers in a well-controlled manner several times in a series. The idea is to make it possible to study the sensorimotor system in a more refined way since it is possible to very precisely control the opening of the discrimination bars, as well as the possibility of designing more complex decision-making experiments. As seen above, studies of tactile discrimination in rodents provide an excellent experimental model to investigate brain functioning. So, we planned a device that favors the animal’s learning and performance, as well as facilitates performing the training for different tasks within the same offered working environment.

The schematic design for the behavioral apparatus is shown in Figure 1. It basically consists of a large main chamber connected to a small discrimination chamber by a 7.5 cm narrow passage. The discrimination chamber is separated from the main chamber by a sliding door that is controlled by the rack system (*Rack and sliding door* in the figure). This rack system is composed of upper and lower racks that are coupled to the upper end of the sliding doors. In addition, it has the bearing base of the doors connected to the motor–gear coupling base where the servo motor is fixed in order to promote the rotation of a toothed pulley, which gives movement to the rack system for opening and closing the sliding door. There are two devices on the front wall of the main chamber for the rodent to insert its nose, labeled *Nose poke left* and *right* in Figure 1. They allow the rodent to signal whether the opening is wide or narrow. Within each of these nose pokes there is a 0.2 cm diameter tube that is inserted into a small water supply hose working as a reward for correct discrimination. Just in front of each nose poke, there is a small metal plate that completely covers it when closed, thus preventing access to the nose poke (not shown in Figure 1). The OCB has an infrared sensor that detects whenever a mouse has inserted its nose into the nasal cavity. The breaking of the light beam sends a signal to the Arduino-based control board that manages the behavioral experiment.

The discrimination chamber contains a third nose poke that is located in the middle of its front wall, shown in Figure 1 and Figure 2. It also has an infrared sensor attached to its side detecting when something has already entered the discrimination compartment. Moreover, there is an IR-based presence sensor in the passage between the main chamber and the discrimination chamber. It is positioned just after the discrimination chamber entrance and has the functionality of detecting the presence of the animal. When the beam of light is broken, the animal is present in the discrimination chamber, so the door will not close. This prevents the animal from being crushed by the sliding door that separates both compartments. On the opposite, when the beam of light is intact the animal is not present. So, the door is immediately closed. This same compartment has two bars made of acrylic (*left bar* and *right bar* in the figures), which are responsible for delimiting an opening between them. Both bars are controlled by the mechanisms called a *scotch yoke*, which, through the rotation of the servo motor coupled to the internal system, transforms the angular movement of the internal pulley of the component into linear movement, thus providing the back-and-forth movements of the bar.

There have two supports for cameras (SC) located at the upper lids of both compartments, labeled *SC-Main* and *SC-Discr* in the figures, for the main chamber and discrimination chamber, respectively. Both supports are made of acrylic and are strategically positioned to capture all the movements or behaviors performed inside each chamber.

At the bottom of the developed behavioral apparatus, we put a structure with a rectangular shape constructed with acrylic, which is the base lid to which the walls of the box are connected. The walls are also constructed with acrylic. A grid of cylindrical bars covers the bottom, totaling an area of 696 cm². This grid is composed of 32 cylindrical aluminum bars 25 cm long and 0.3 cm in diameter. The tray is built-in, easy to insert, and made of acrylic. The apparatus may be placed inside a soundproof and lightproof insulation box. The basic modules roughly described here will be detailed next.

### 3.2. Base Platform and Modules

In order to make the physical structure parts, we use materials and ready-made elements such as sensors and actuators. The massive structure of the developed device is fabricated with acrylic and aluminum, composed of several pieces or parts such as 1 acrylic base, 15 acrylic walls, 2 extra acrylic insert bars, and 32 solid aluminum 0.3 cm cylindrical bars. The thickness of acrylic sheets is 0.5 cm, which makes it possible for the acrylic to be used even with withstanding hydrostatic pressure. With a low thickness, the acrylic provides very light pieces facilitating their transport, which makes it ideal for fabricating smaller items for the developed device. Besides being very flexible materials that can form different shapes in different sizes, acrylic sheets usually have a minimum durability of 10 years, resisting any type of climatic action. Thus, the construction of structural components with this material was our choice. The cylindrical bars are modular structures made of aluminum, intended for covering the base, functioning as the floor of the main chamber. As described above, it took 32 pieces with diameters of 0.3 cm and lengths of 25 cm, which, together, form a grid or a set of bars.

Some device-specific modules can be found on the market, ready-made, among which we highlight three nose pokes, two scotch yokes, one rack set, one reward bomb, and two camera supports. However, in order to integrate these modules at this stage of the project, it was necessary to develop some customized parts. For this, we use 3D prototyping that allows the manufacture of parts more suitable for the project, increasing the predictability of errors and ensuring that the device is built faster, as a whole, since it is not necessary to search or to wait for alternatives in the market.

The possibility of printing very complex parts at once, with the exact amount of raw material needed for their manufacture, is one of the great advantages of choosing 3D printing in the manufacture of some parts. With this technology, modular components such as the nose poke are easily coupled to the project. The internal parts of the scotch yoke and rack set were printed in acrylic. The choice for the development of these parts in 3D printing favored a considerable reduction of costs, avoiding waste. In addition, the high precision in the design allows the perfect fit of replacement and maintenance parts, saving time with adjustments and corrections. Our final product developed has confirmed the agility and versatility of this process and the excellent cost/benefit ratio in choosing this solution.

### 3.3. Electronic Materials and Components

The requirements always change during the development of a product and, as the requirements engineering process developed by everyone in the project improves, problems found are smaller due to all the difficulty involved in this important part of the analysis. With this in mind, here we put the various materials and equipment used in the construction of the box, proposing an update or an upgrade of older ones, as mentioned above. In this section, we try to describe the equipment and materials, aiming to understand the platform as a whole.

Among the device control materials and equipment are motors, microcontrollers, and sensors, whose main characteristics are described below. As seen above, we use the Arduino At Mega as a microcontroller. In the motors category, in the project, three Mg995 Servo Motors, two sg90 Micro Servo Motors, and two peristaltic dosing pumps are used. As for the only type of sensor used, for now, there are three pairs of IR Breakbeams, discussed further.

Regarding the Arduino Mega, in this project, we used 13 PWM ports, 28 digital ports, and 6 analog ports to fully control the behavioral box. Additionally, we need a 64-byte buffer, and to efficiently process at least one processor with flash memory of 256 KB (8 KB used in boot loader), 8 KB SRAM, 4 KB EEPROM, and 16 MHz clock speed. We also need support for an SD card bed where the routines should be stored. In view of these needs, the Arduino Mega 2560 R3 controller board was our choice, due both to its cost (less than USD 30) and its robustness and small size.

#### 3.3.1. Actuators (Motors and Servo Motors)

The Micro Servo 9 g SG90 is used to act on the *nose pokes* because its dimensions fit the intended parts and because it is a high-quality servo and excellent in robotics projects with Arduino, PIC, Raspberry, and others. This servo is suitable for projects such as RC helicopters and branded planes such as Hitec, Futaba GWZ, and JR. In a way, it ends up being integrated into our mechanism to control an access bar to the nose poke. It has a rotation angle of 180 degrees and comes with a three-pin cable for power/control and various accessories. The Micro Servo 9 g SG90 is an excellent alternative, due to its cost (less than USD 5) and due to its robustness and small size. It has the following specifications: operating voltage: 3.0–7.2 V; rotation angle: 180 degrees; speed: 0.12 s/60 degrees (4.8 V) no load; torque: 1.2 kg·cm (4.8 V) and 1.6 kg·cm (6.0 V); operating temperature: −30∼+60 °C; gear type: nylon; cable size: 245 mm; dimensions: 32 × 30 × 12 mm; weight: 9 g.

The use of the MG995 engine has as its main characteristics high torque and resistance. It is an essential component for robotics, mechatronics, and many other projects. In robotics, the servo motor is responsible for moving robot arms, legs, and hands. In model car racing, the servo motor is used to turn the front wheels of the carts, and in model aircraft, it is used to control the flaps of airplane wings. In this project, its function is commanding the opening of doors and bars. MG995 Servo Motor connections are compatible with Futaba, JR, Hitec, GWS, Cirrus, Blue Bird, Blue Arrow, Corona, Berg, and Spektrum, among other standards. It has metal gears and it has a torque of 9.4 kg/cm at 4.8 VDC and 11 kg/cm at 6 VDC. In addition, this servo can rotate up to 180${}^{\circ }$.

The peristaltic dosing pump (from Intllab) is widely used in the field of experimental analysis, biochemistry, pharmaceutical, fine chemicals, biotechnology, pharmaceutical, products, ceramics, water treatment, environmental protection, etc. In the current project, its role is to provide a reward for the nose pokes.

#### 3.3.2. Infrared Sensors

In this project, infrared (IR) sensors are a simple way to detect motion. Here we use access monitoring mechanisms to routines and compartments of the behavioral box. They are presently integrated into the nose reward pokes and the main one. They work by having an emitter side that sends out a beam of invisible human infrared light, then a midway receiver that is sensitive to that same light. When something passes between the two and is not transparent to the IR, then the *beam is broken* and the receiver will warn the user.

Compared to PIR sensors, break beams are faster and allow for better control of where the user wants to detect motion. Compared to sonar modules, they are less expensive. However, it needs both the sender and receiver on opposite sides of the area in order for it to be monitored. It has the specifications shown in Table 1.

### 3.4. Behavioral Box Functioning Logic and Control Software

Figure 3 shows an illustrative example of the basic working logic of the behavioral box, in two steps, with four actions in the second step. Right after the code is loaded (step 1), it is observed that the modules in the chambers are enabled (green stripes) and others deactivated (red stripes). Task 1 can be executed and then, the activation of each next task can be concatenated (arrows 2, 3, and 4), making some proposed training run correctly. In the example, some low-level control tasks of the behavioral box would be performed. The necessary actions in Figure 3 can be performed by the Arduino Mega microcontroller [12]. As mentioned above, with this functioning it is possible to design complex experiments. For example, it is possible to have smaller variations in the opening of the bars and study how these little variations behave in the rat’s brain firing. This functioning will be better explained next, as well as the software part necessary for it.

Figure 4 shows the basic functioning, which is divided into three phases. First, the Arduino sends commands to the scotch yoke module, which orders the opening of the bars. Then, the rack module carries out the command to open the door sliders. The nose central poke is then activated completing the first stage of preparation for use. After being ready for use in the previous step, the system waits for the Arduino, which will give the command to the scotch yoke module that will open the bars, in narrow or wide mode. Then the rack module will open the sliding door releasing access of the rodent to the discrimination chamber, which is detected by the nose central poke that is activated when it passes in front of the infrared sensor.

In the sequence, the Arduino activates the nose reward poke for a set time, and simultaneously the rack module closes the sliding door. In case the nose poke is correct, the infrared sensor’s light beam break will be detected and the reward pump module will be activated, with the mini servo releasing the access bar for reward reception. Nonetheless, if the nose poke is wrong, the mini-servo does not release the access bar for reward. After all of these steps, the Arduino will restart the process coming to the first step, and eventually start a new training or attempt.

We use Python for implementing the operating logic. All of the mechatronics are left in slave mode when the software starts to control, sending all the command steps. This means that the electronic circuit associated with the Arduino hardware becomes a complementary element to the operational functionalities of the device. We notice that we have developed a new behavioral apparatus with several new resources. This new setup has a newly developed code for the several threads that were implemented in the Arduino IDE and paired with software in Python. The supervisory system is connected to the Arduino via USB communication, with the function of adding or locating any hardware, as shown in Figure 5 and Figure 6.

### 3.5. Development of Software Tools

All actuators and sensors are controlled by the Arduino hardware in an integrated way, thus enclosing our mechatronic system. As said above, the entire system controlled by Arduino is left as a “slave” of the “master” control software developed in Python. This requires (and enables) the development of software tools in order to process the experimental tasks with rodents, facilitating and making it user-friendly with an interface. The software tools comprise two parts, the control itself and the supervisory system.

As said, the control part is developed in Python, which is a high-level, scripting, imperative, object-oriented, functional, dynamically typed, and strong programming language [13]. In order to perform functional tests and calibrations of the functions performed by the behavioral box, we developed the executable routine shown in Figure 5, which is executed before each training process in the behavioral box.

To start this step, one must choose the serial port to Arduino and then connect, using the options shown in the top-left part of Figure 5. This control version also has control commands (e.g., *Open* and *Close*) for all the devices (e.g., water release) and/or doors (or bars) involved in the functional steps of the box, namely:In the main chamber: the nose sensors and poke actuators and the timing of reward release in milliseconds.The main door that gives access from one chamber to another.In the discrimination chamber: the opening of the bars and their possible variations of openings (narrow, wide, and totally open).

The sensors are represented by green (“on”) and red ( “off”) on the control system screen (Figure 5).

#### 3.5.1. Behavioral Box Supervisory Software

Supervisory systems are widely used for supervision, operation, and control, normally in industrial or similar applications. In this work, as shown in Figure 6, we implement the supervision software as a single executable routine, which contains the real-time monitoring functions of the tasks being performed inside the behavioral box according to a given experiment purpose.

In this supervisory screen, the user has at the top of the screen the *Menu* of software functions with the main commands (*File*, *Manage*, *Results*, and *Help*). Then, it displays the option for selecting and connecting with the control hardware (Arduino) and some buttons for verifying the functional status of the sensors with the option to select them as *on* or *off*, as shown in Figure 6.

One can see the *Update* button for the cameras, which makes it possible to watch and record the workouts. The camera image will be displayed in the black square spaces shown, where the frames where the cameras of each area of the behavioral device are located will appear. One side will record the actions happening in the main chamber, which is where the rewards are provided. The other camera will record the actions in the discrimination chamber, which is where the center bars and nose pokes are.

Below these registration screens, we have the animal selection buttons (rodents) and the type of training that will be performed. Then we have a presentation box with information about the activity that will be executed, showing the type of discrimination, success/correctness in the execution, time, and status. Next to these features, we have the buttons for executing the scheduled tasks, as follows. The status of the experiment (*Running*/*Not Running*), the starting of the experiment, and the success count (or correct executions) on each side of the reward, as well as the total number of hits. Finally, at the bottom of the screen, we have the progress bar of the training being operated in the behavioral box, as shown in Figure 6.

#### 3.5.2. Supervisory Screen Menu

When the user uses the icons at the top of the screen *Menu*, activating the main commands (*File*, *Manage*, *Results*, and *Help*), then selecting the *Manage* button, it shows the options of *Animal* and *Training Program*. As shown in Figure 7, information about the animal can be added in this section, such as name and weight, among others. Figure 8 shows the selection of the *Manage* button and the *Training Program* icon, where the types of activities to be performed and the time for each of the steps will be added.

The information from the training performed for a given registered animal appears when the user uses the icon at the top of the *Menu* screen, activating the *Results* command, as shown in Figure 9.

Supervisory systems generally have the functionality of being integrated with a database and thus enable historical records of the general data processing performed. In addition, they have tools for generating various reports, whether textual, graphic, or in a mixed format. Depending on the system’s implementation architecture, the reports can be made available at the process’s operating station, on other machines on the network, or even via the Internet. The supervisory system connects with process equipment (PLCs and/or other devices) through specific communication drivers. Here we have the Arduino, which allows communication through the most diverse existing protocols. This characteristic makes the supervisory systems widely used as a source of process data for other applications, as shown in Figure 9. Therefore, these are the main functional characteristics of the software developed in order to meet the tasks performed in the internal environment of the box before, during, and after experiments.

### 3.6. Integrated View of the Complete Behavioral Box

Following the design of all the components of the box and the survey of the necessary materials and modules, the planning of the training of the rodents was carried out, to be worked as tasks, and the automation of the complete box itself. After this design phase, the analysis of the suitability of materials was carried out, which will be discussed in this section through a complete view of the device, as shown in Figure 10. It should work fully integrated with Arduino, which is the system brain that takes care of all functionalities. We notice that at this time some changes were still possible (and necessary), such as the planned syringe pump module that appears in the project images in the sketch shown in Figure 10 that was replaced by peristaltic pumps. This modification brought greater precision in the provision of rewards and aesthetically streamlined the final project.

The mechanisms of movement of the bars, the release and delivery of rewards, and the opening of sliding doors work in an integrated way and together they collaborate so that the training can be carried out easily. Notice that the training footage is obtained simultaneously with the training process of each animal.

The door for inserting the animal into the box is much larger, thus facilitating the handling of materials such as the box in which the animal is transported, thus favoring the user to have more freedom of work. The walls and internal spaces are faithfully maintained with their dimensions and positioned angles. Additionally, its base has a favorable layout for possible cleaning and structural maintenance, as shown in Figure 11.

The reward nose pokes shown in Figure 12 are integrated with the peristaltic pumps, which were designed to act precisely in the release of the reward, as shown in Figure 13 and Figure 14.

The movement of the discrimination bars is linked to the scotch yoke that acts in determining the space of a narrow (62 mm: left column) or wide (68 mm: right column) opening. The sliding door is controlled by the rack-top system, which allows the rodent to have access from the main chamber to the discrimination chamber. In addition, the equipment has two spaces on its upper part, as shown in Figure 14. This is to allow the insertion of solid bars, for performing different training experiments from what it has as its main training (as illustrated above in Figure 3).

Closing the construction part of the CCO, all the structural parts of the device have been mounted with the acrylic and 3D printing parts. The acrylic parts can be seen in Figure 15 and Figure 16. The modular parts acquired (including sensors and actuators) were fixed to the structural part of the box, as well as the integration of the physical mechanisms with the computational and control elements, and then we proceeded with experiments to validate the behavioral apparatus. The final version is shown in Figure 17 (final hardware) and Figure 18 (final software working on the computer screen).

## 4. Experiments and Results

Our main purpose is to demonstrate the functional validation of the behavioral apparatus developed and to show the different types of training that are possible to execute in the box, besides the monitoring of the training steps by the sensors. In addition to effectively implementing these process improvements, making the new tool available to the community. Thus, we envisioned two alternative ways to validate our proposal. The first would be through using live animals for performing the training, which will not be performed in the current work, as the goal here is only to validate the correct operation of the proposed enhanced device. The second option is the one that we have adopted here, through continuous and repetitive tests using a remote control mechanical mouse model. We notice that this is a good choice because we do not need at this development time to sacrifice the animals after the tests (which appears to be mandatory in such types of experiments) and this option does not need to have its ethics protocol formalized as it does not involve live beings. As it will be shown, there are some new types of training that could not be carried out with old methods, but that can be done with the changes and using the new apparatus proposed in this work. Below we highlight the whole set of training that can be performed with this new model.

### 4.1. Training Validation

Basically, an experiment consists of placing the electronic mouse in a given position inside the behavioral box. In this case, it is put on the front of the wall of the chamber where there are two nose poke mechanisms for the rodent to insert the nose. As said above, these mechanisms allow the rodent to respond if the width of the discrimination chamber is wide or narrow. The training consists of the discrimination of the width of the bars for each reward. Inside each of these mechanisms, there is a 2 mm diameter tube into which a small water supply hose is inserted, being the reward for correct discrimination. Directly in front, there is a small ABS plate that completely covers it when it is closed, thus preventing improper access.

The sensor is activated whenever the animal puts its nose in the nose poke and activates an infrared sensor that detects that the mouse has inserted its nose to respond. When the light beam breaks, a signal is sent to the Arduino that controls the behavioral experiment. The discrimination chamber contains a third sensor rig located in the middle of the front wall that also has an infrared sensor attached to its side, which detects when something has entered the discrimination compartment.

When passing through this compartment, the animal will leave the discrimination chamber and the main door with access from one chamber to another will be closed. When the width of the bars is narrow, the mouse will receive the reward on the left side and on the right side when the width of the bars is wide. The two accesses to rewards are opened simultaneously, providing a drop equivalent to 25 micro-liters of water if the mouse gets it right. If it misses, it will not receive the reward, so the stage is restarted again, closing the access to the rewards and opening the main door of access to the discrimination chamber. Thus, a new random draw is made to open the widths of the bars. In the case of live animals, at the end of the process, they would be returned to the vivarium for use in other activities or may be sacrificed if not used anymore, depending on the ethical protocol, which is not the case here.

### 4.2. Learning to Decide (Validating the Proposal)

As mentioned, the behavioral box developed so far has new technological and functional aspects. Technologically, we can highlight the fact that it is a modular device, that is, any change of parts or compartments can be carried out without compromising the other modular structures. Functionally, the behavioral apparatus has two inputs to insert extra bars for forced training. With this barrier insertion, a new behavioral protocol can be easily developed.

Thus, the model created enables the discrimination training described in the functional items of the box, with the main issue being the training itself. In addition, it can enable forced training with bar discrimination and receive forced rewards (left or right). This allows for carrying out a directive forced training, without the need to receive a reward only with the discrimination of the bars.

In general, we can divide the training into two phases or stages. In the first, the animal performs the behavioral task with the physical barrier blocking the right side, and, in the second stage, blocking the left side, until they reach the learning criterion. Through the behavioral box developed, we can fully monitor the training performed by the rodents, as well as control in a more refined way the opening of the discrimination bars.

#### Experimental Results with Robotic Mouse

We use an electronic mouse that can move forward and backward using a remote control, as shown in Figure 19. It is made of plastic in the colors gray, black, or brown. The power supply includes 2 × 1.5 V AAA batteries. Its weight is 100 g and its size is 22 × 7.5 × 6 cm.

These validation tests were carried out in the vivarium of the Santos Dumont Institute (ISD) located in Macaíba, Brazil, with the rodent model shown in Figure 20. This experiment will serve as a test of the sensory elements of capturing movements of the box, in addition to performing image capture tasks.

A behavioral task has been devised that requires the rodents to discriminate the width of a wide or narrow opening using only their large mystacial vibrissae. For this, the sensors and actuators must be precisely able to capture any activity coming from inside the box. The monitoring of the task is carried out by the supervisory software developed for this study, through two cameras, located one in the main chamber and the other in the discrimination chamber, as shown in Figure 21 and Figure 22.

The mechanical mouse used in this test training quickly served this task and could be a tool for accurately simulating the discrimination between bar-width openings and activating the sensors in the central nose poke with the nose. Precise discrimination required a large number of hits in the holes of the reward and discrimination nose pokes, which were important for the success of the motor task performed. Then, the movements inside the behavioral box were executed a large number of times, in order to obtain the greatest number of possible operational failures during the tasks.

In this training, we evaluated the learning conditions in the internal environment offered by the box, through image and video verification. We can record the training and perform a post-analysis of the performance of real rats that can be subjected to training in this behavioral box. As the mice must pass through the nose pokes, this test task confirms that this box has enough tools to offer a good learning environment. In all tests made, the sensors have been activated and the rewards are released. One of the tests is shown in Figure 23 and Figure 24.

### 4.3. Discussion

As seen above, for carrying out experiments in neuroscience, apparatuses are normally used in which the rodent performs some task to receive a reward, known as behavioral boxes. Rodents are known to be an excellent model to study the tactile sensory system, as these animals use their vibrissae to locate and discriminate objects. The apparatus proposed in this work has been initially validated for these tasks. The next step is to use real rodents from our university lab in order to allow research in this area to be done without the dependence on commercial houses, thus making it easier to work.

The goal here was to develop the box, enhancing existing structures in our lab. To this end, we have accomplished this goal. Thus, further experimental work involving studying and understanding animal behavior can be done, as will be suggested next in the conclusion section of this work.

All aspects of the training were fully automated and controlled by the Arduino, requiring no experimenter intervention. Operation is controlled by the Arduino IDE programming software. If the behavioral apparatus is enclosed in a lightproof isolation box, it can be illuminated with infrared light. In this case, the behavior of each mouse can be monitored via infrared-sensitive video cameras (placed above the discrimination chamber and main camera on their supports), thereby eliminating any need to obscure the mice’s view.

In such a way, the construction of equipment composed of several simple replacement mechanisms provides significant relevance in the handling and operation of the device. Modularity is also a plus in the current project, allowing for extra components related to some given mechanism or experiment. This offers extra features such as two bars to insert forced training in the main chamber, a pair of reward nose pokes, the possibility of taking the device to environments outside the laboratory, and even the adaptation of a small camera to the device. The modular components can be combined with each other virtually unlimitedly. In this way, the ideas of obtaining multiple functionalities become a reality. Reliability and expandability ensure that the built modular systems can be used over a long period of time.

## 5. Conclusions

In this work we provide improvements on the devices used to study the tactile sensory system of rodents by designing and building a new model of a behavioral box in which the animals have to discriminate the width of an opening, using only their vibrissae to receive the reward.

The behavioral study device developed is an important tool for the study of the tactile sensory system. The work developed here allows other laboratories to use this platform in an accessible and easy-to-develop manner. All the developed code is available on the GitHub multi-user collaborative platform (github.com/renanaraujoeng/behavior_box, accessed on 21 December 2022) and all of the documents and pictures can be retrieved from our site (natalnet.br/sibgra-pi2022/behavioral-box, accessed on 21 December 2022). It has been initially validated and is ready for further use.

From now on, the box will be used in research with real rodents, for which we plan to run behavioral experiments in order better verify how much the developed box improves in the data acquisition and analysis of these experiments. Other developed computer vision tasks for the experiments that are being developed in parallel to this project [14] will also be added to the current set of tools, which will even improve the current set. An ethics protocol has been elaborated and is been verified by our organisms in order to perform that new set of experiments.

## Figures and Tables

**Figure 1 sensors-23-00288-f001:**
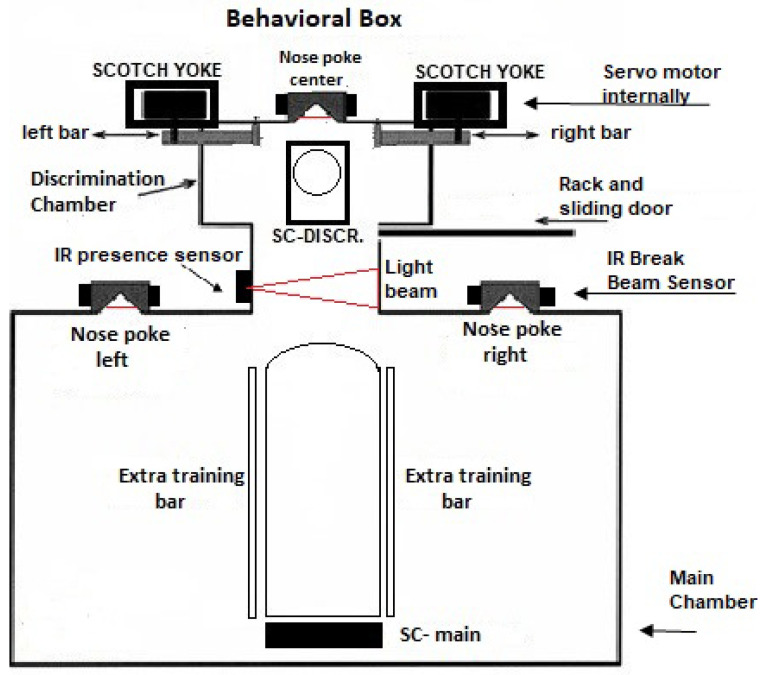
Schematic design of the behavioral box.

**Figure 2 sensors-23-00288-f002:**
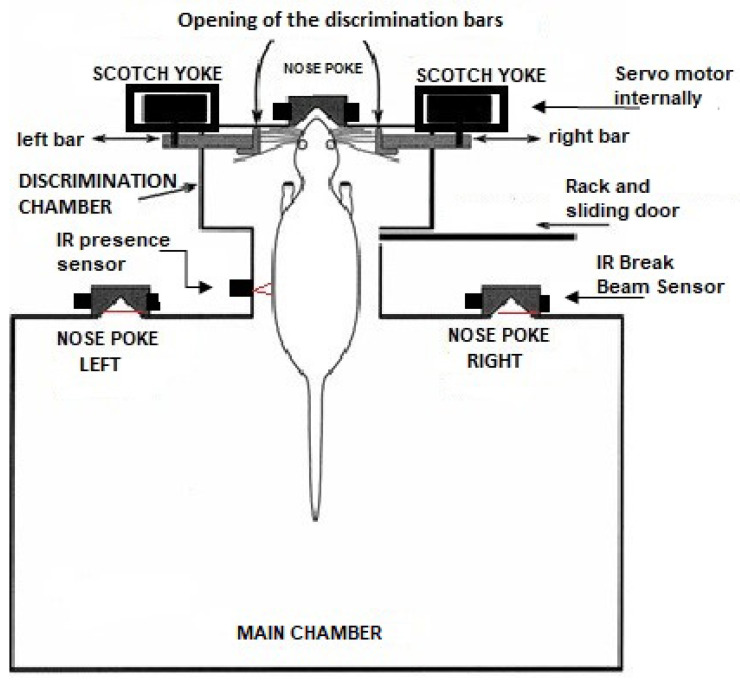
Behavioral box schematic without the extra training bars, with a rat in the discrimination chamber illustrating its functioning.

**Figure 3 sensors-23-00288-f003:**
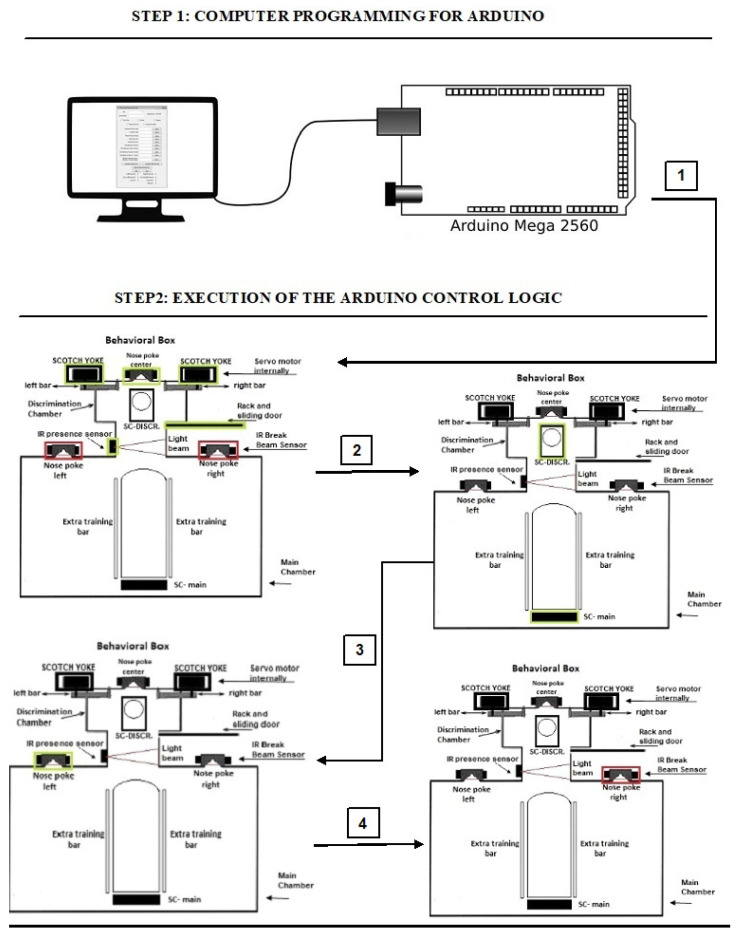
Working logic of the behavioral box.

**Figure 4 sensors-23-00288-f004:**
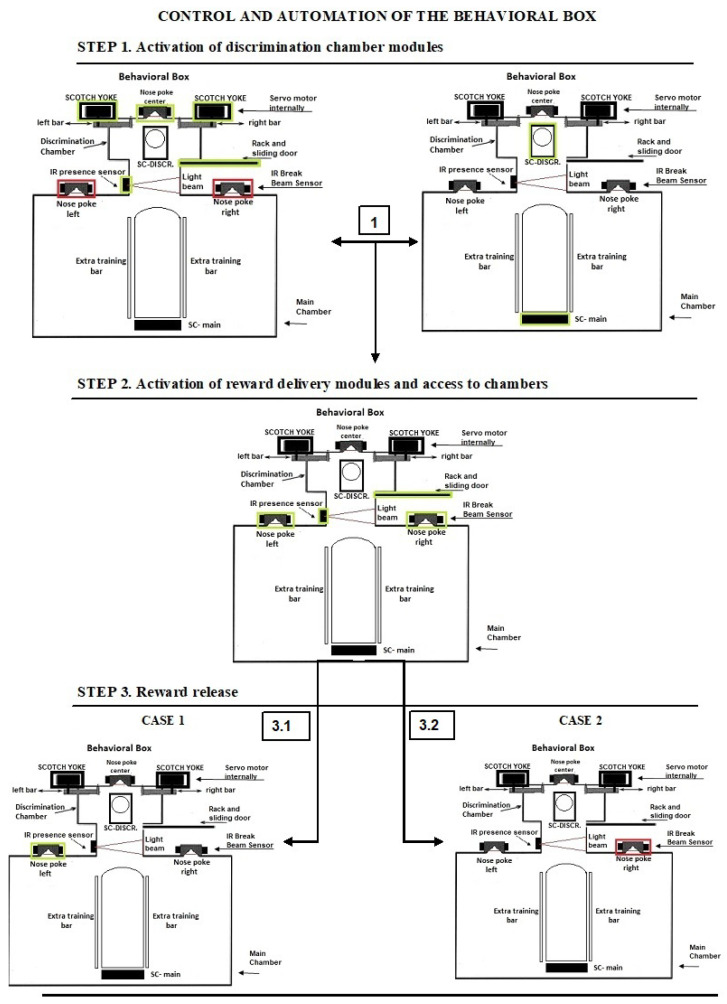
Basic functioning diagram, with three phases and two sub-cases in the last phase (3.1 and 3.2), which are just examples of some processing. In 3.1 the nose poke left is turned on and in 3.2 the nose poke right is turned off.

**Figure 5 sensors-23-00288-f005:**
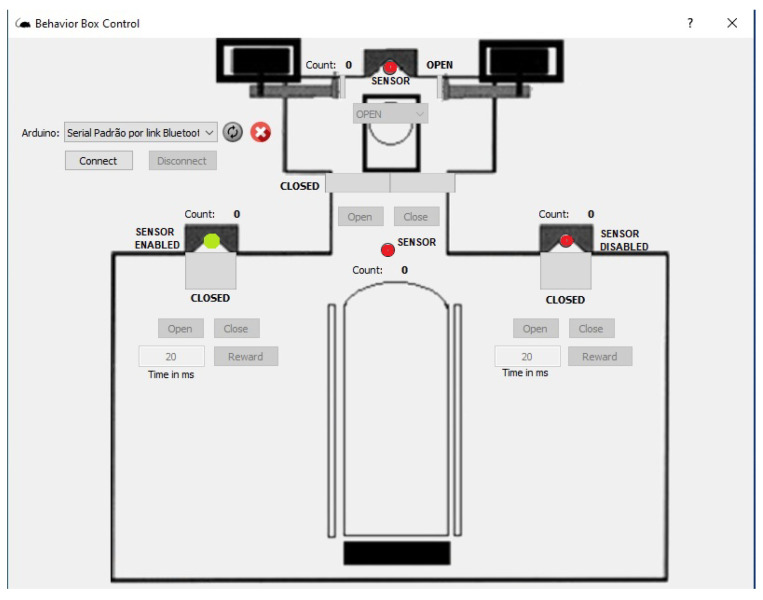
Behavioral box control software interface.

**Figure 6 sensors-23-00288-f006:**
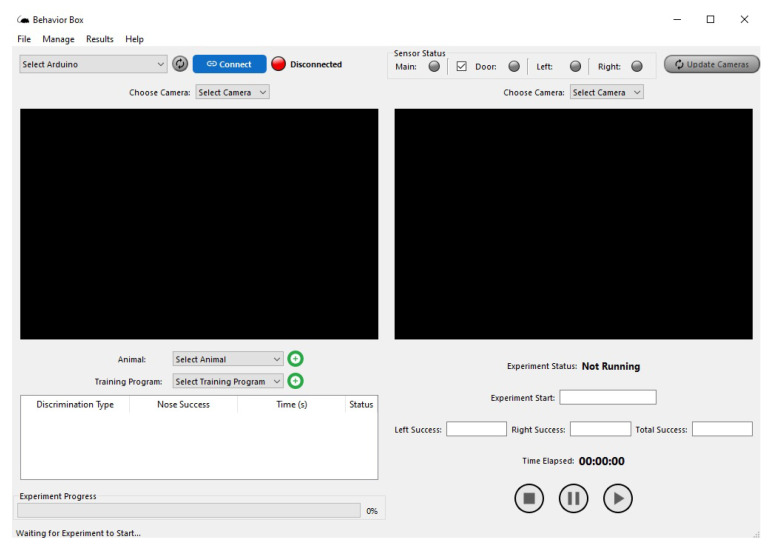
Software: behavioral box supervisory screen.

**Figure 7 sensors-23-00288-f007:**
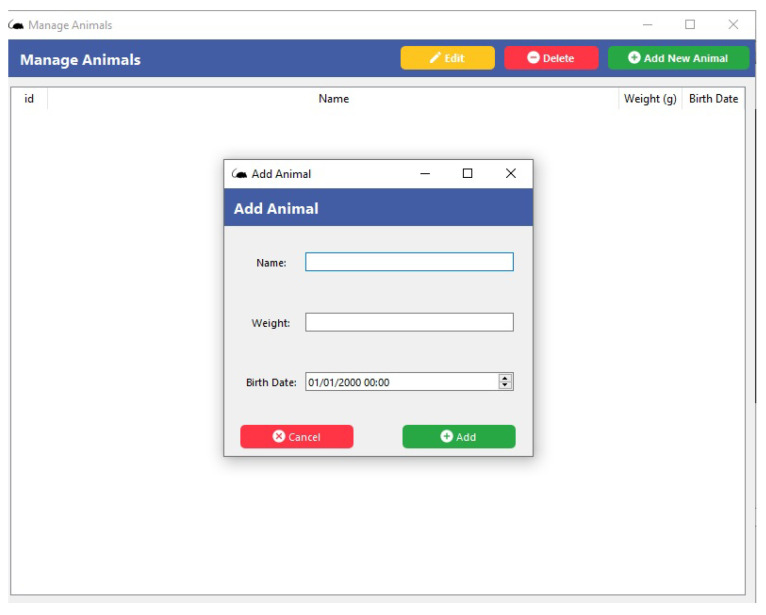
Screen of the *Animal* function of the supervisory software of the behavioral box.

**Figure 8 sensors-23-00288-f008:**
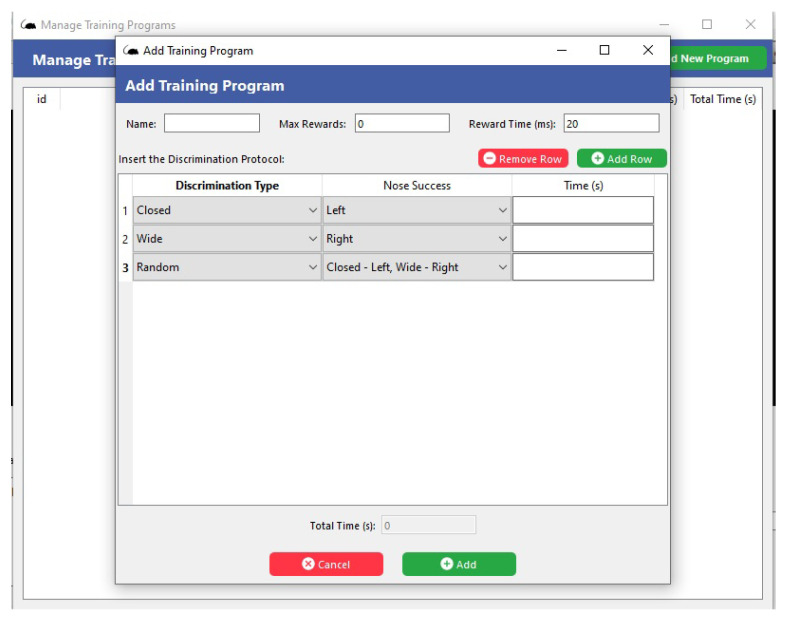
Screen of the *Training Program* function of the supervisory software.

**Figure 9 sensors-23-00288-f009:**
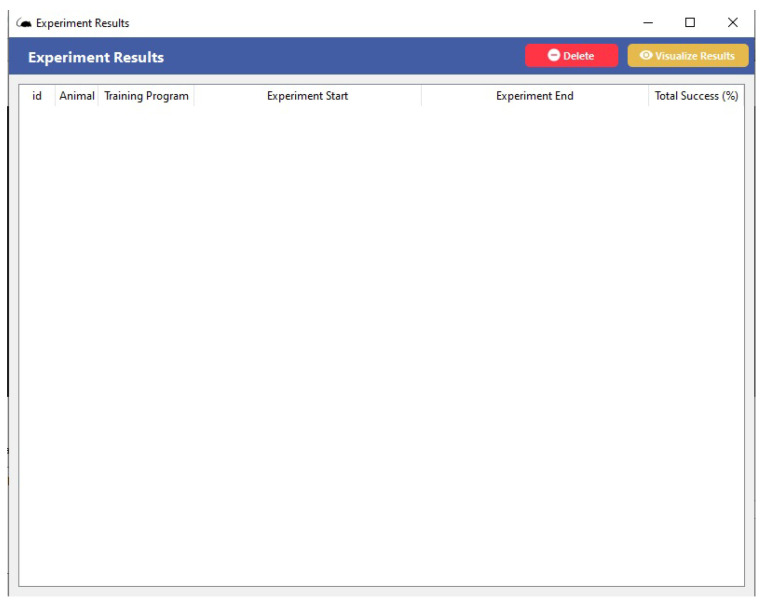
Function screen *Results* of the supervisory software of the behavioral box.

**Figure 10 sensors-23-00288-f010:**
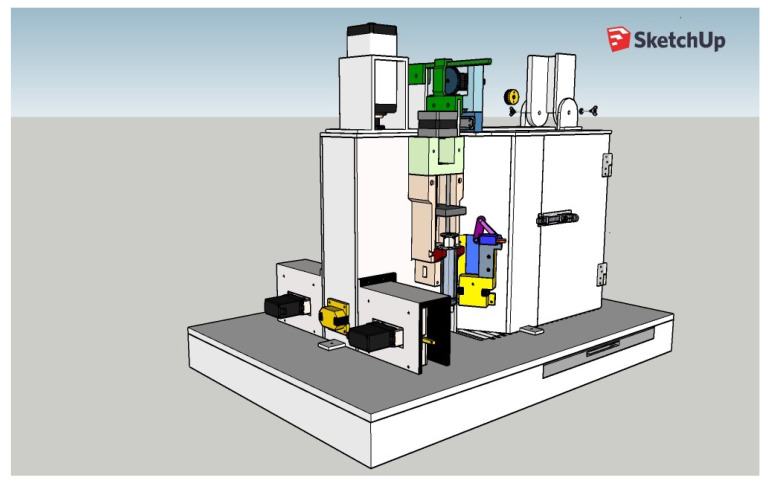
Sketch (project) of the complete automated behavior box.

**Figure 11 sensors-23-00288-f011:**
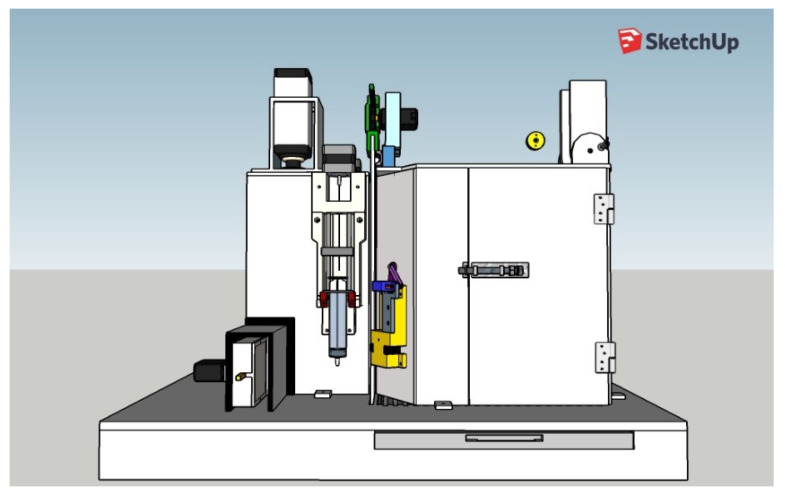
Full automated behavior box: side view.

**Figure 12 sensors-23-00288-f012:**
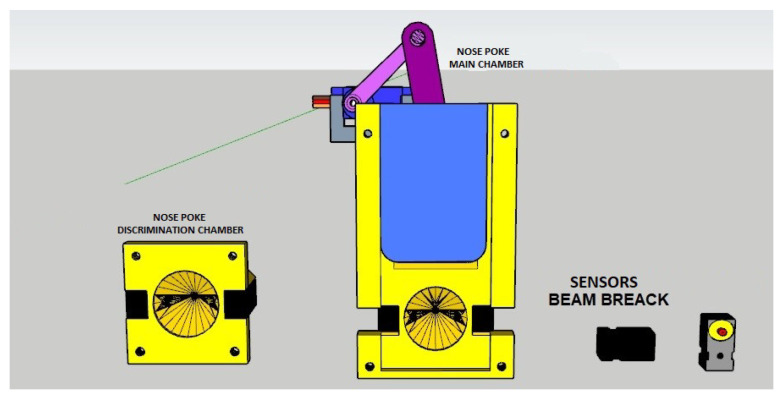
Nose poke front view.

**Figure 13 sensors-23-00288-f013:**
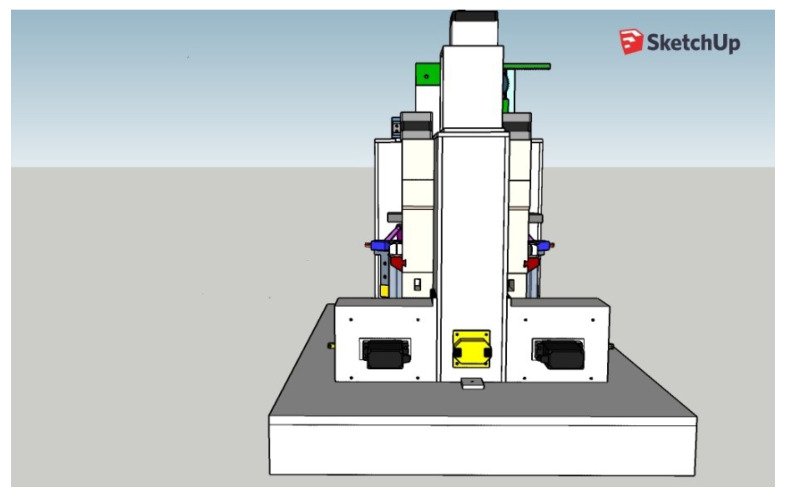
Full automated behavior box: front view.

**Figure 14 sensors-23-00288-f014:**
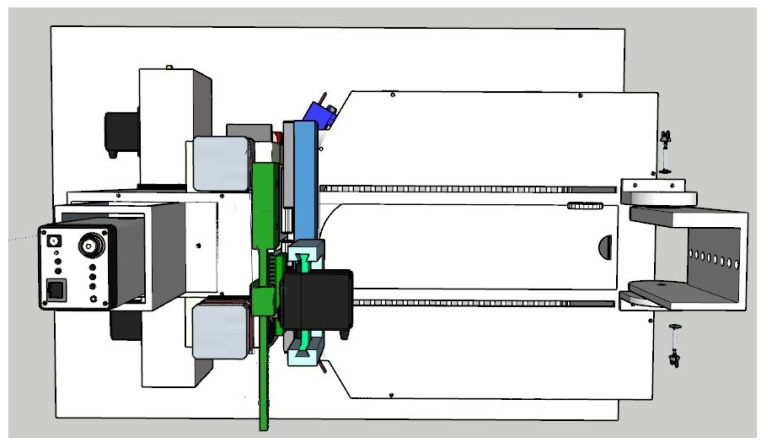
Full automated behavior box: top view.

**Figure 15 sensors-23-00288-f015:**
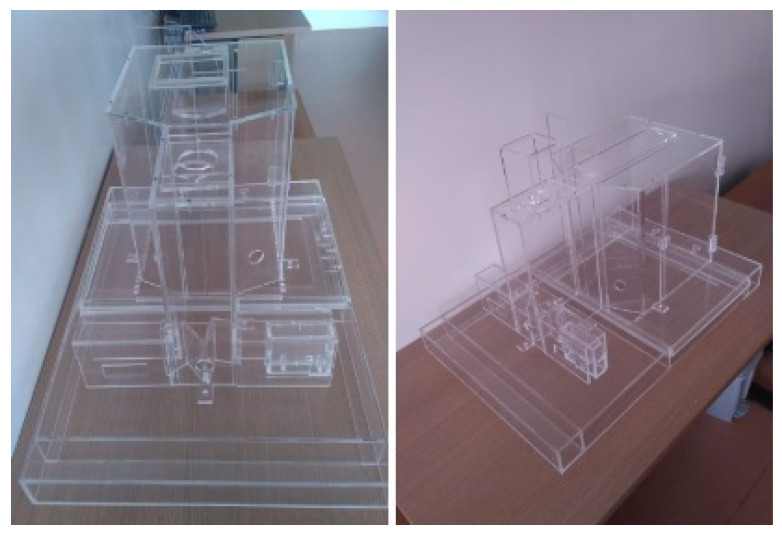
Final assembled behavioral box in front and oblique views.

**Figure 16 sensors-23-00288-f016:**
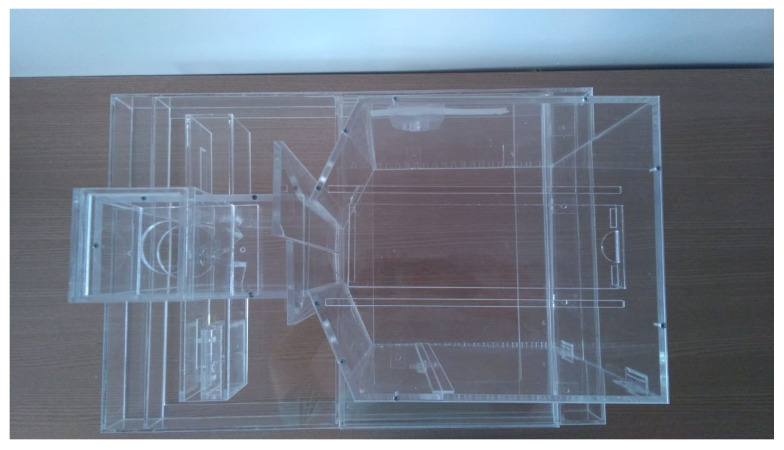
Behavior box top view.

**Figure 17 sensors-23-00288-f017:**
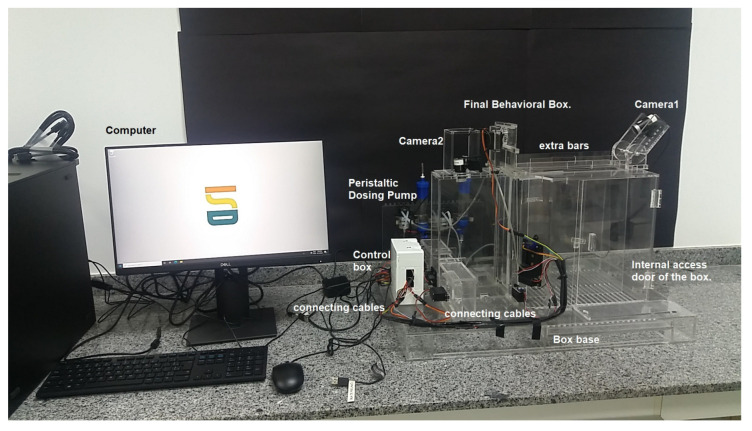
Behavior box working—final version.

**Figure 18 sensors-23-00288-f018:**
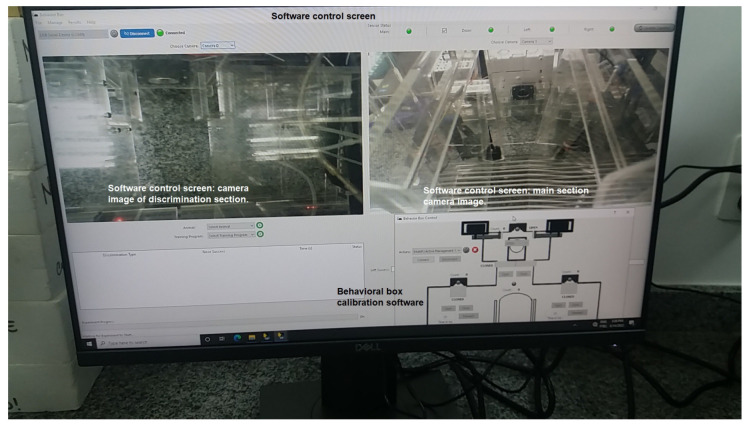
Behavioral box: software—final version.

**Figure 19 sensors-23-00288-f019:**
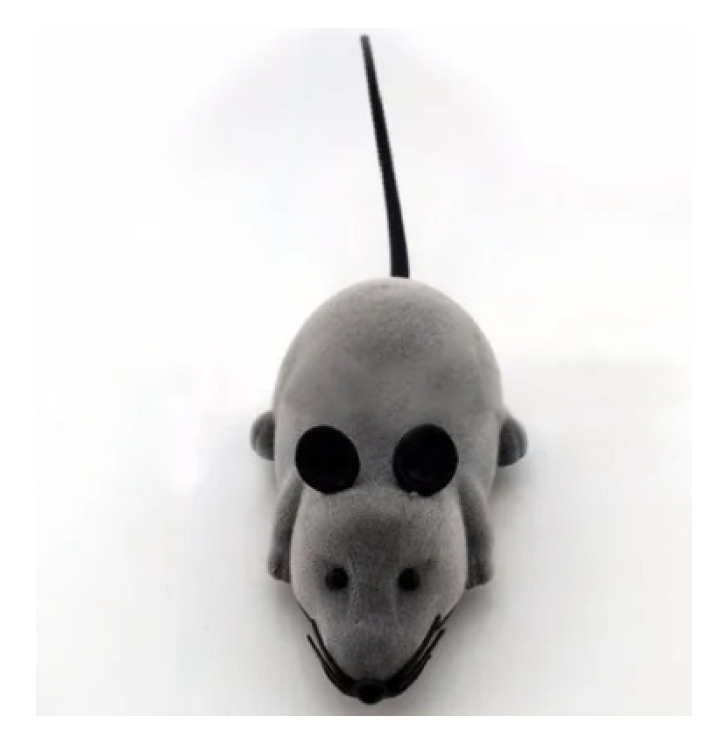
Remote control mouse.

**Figure 20 sensors-23-00288-f020:**
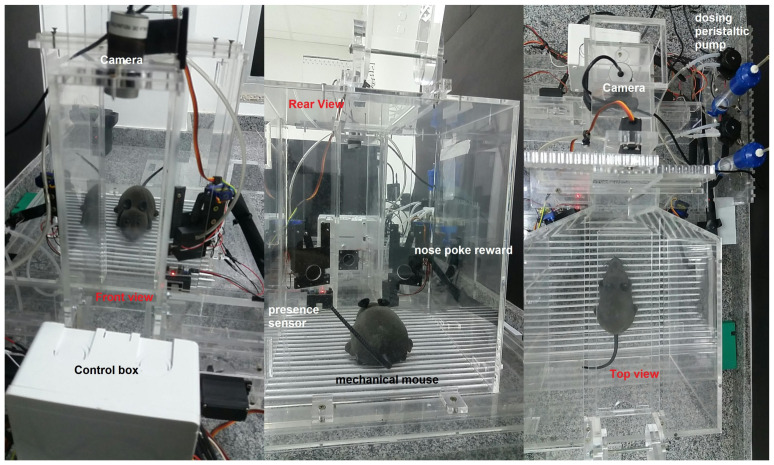
Remote control mouse inside the behavior box.

**Figure 21 sensors-23-00288-f021:**
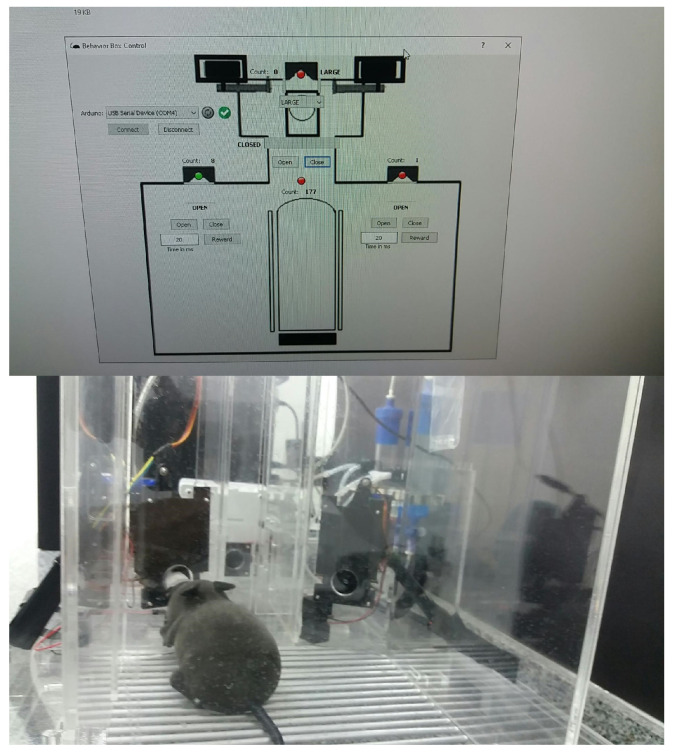
Remote control mouse in behavior box: activating the left reward sensor, referring to the wide position of the bar.

**Figure 22 sensors-23-00288-f022:**
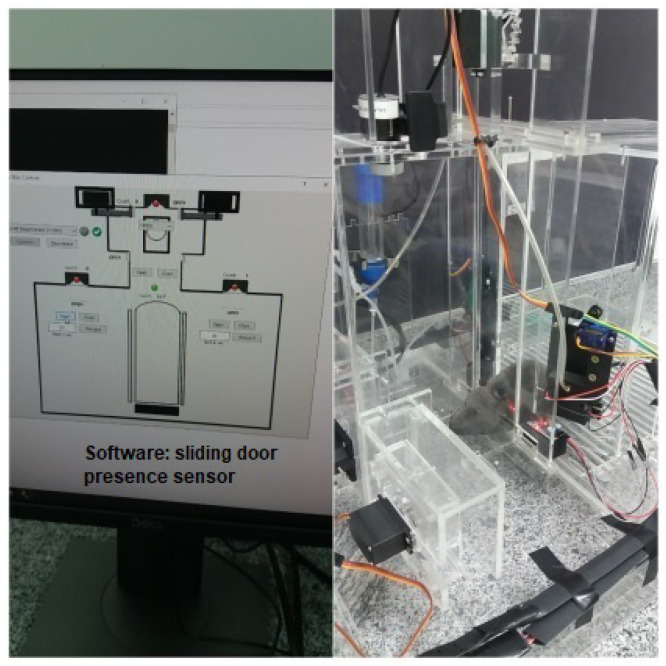
Remote control mouse in behavior box: activating the central sensor, for mouse safety, preventing crushing.

**Figure 23 sensors-23-00288-f023:**
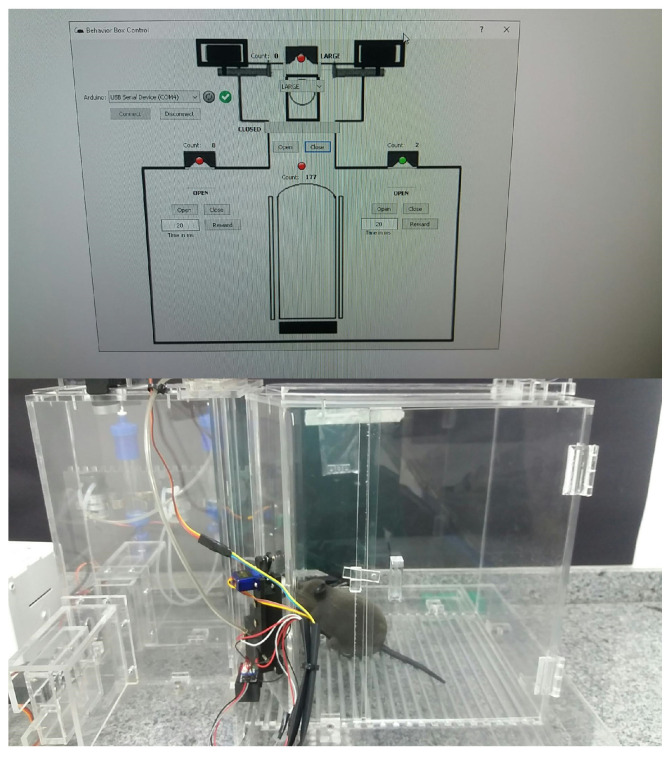
Remote control mouse in behavior box: activating the right reward sensor, referring to the narrow position of the bar.

**Figure 24 sensors-23-00288-f024:**
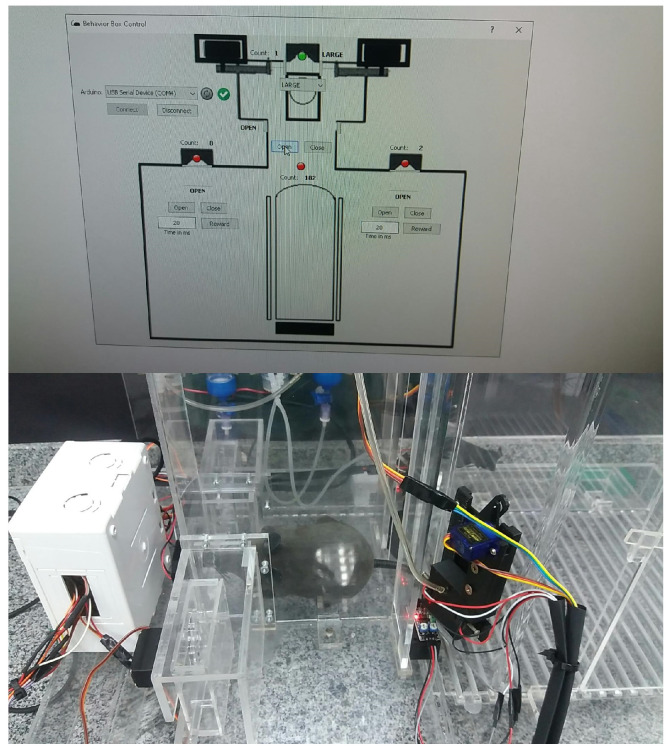
Remote control mouse in behavior box: activating the central nose poke sensor, referring to the discrimination step, with the mouse vibrissae touching the bars.

**Table 1 sensors-23-00288-t001:** Specifications of the break beam used.

Feature	Value
Detection distance	approx 25 cm/10″
Supply voltage	3.3–5.5 VDC
Emission current	10 mA to 3.3 V, 20 mA to 5 V
Receiver output current capacity	heatsink 100 mA
LED angle from Transmitter/Receiver	10${}^{\circ }$
Response time	less than 2 ms
Dimensions	20 mm × 10 mm × 8 mm/0.8″ × 0.4″ × 0.3″
Cable length	234 mm/9.2″
Weight (of each half)	approx 3 g

## Data Availability

Not applicable.

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
