# Peer review of "Improved Behavioral Box and Sensing Techniques for Analysis of Tactile Discrimination Tasks in Rodents"

_sensors, 2022, doi:10.3390/s23010288_

Round 1

Reviewer 1 Report

The paper presents a system (Behavioral Box) to perform studies with rodents. The main contribution is that the system is built with low cost available resources. Although the work is valuable, it is not a research paper in my opinion (“original research manuscripts, provided that the work reports scientifically sound experiments and provides a substantial amount of new information”). It could be reported as a Technical Note and details of the project (schematics, code, dimensions, files for 3D printing, etc) should be provided to ease its reproduction.

As a general comment, I would say that the paper is verbose, many information is already known in the context of the Sensors Journal and should be removed (for instance long descriptions of Arduino, Python, etc.). There are also other parts related to the context the system is aimed for that should be summarized. The paper should focus on the developed system mainly.

Regarding the description of the Behavioral Box (which is the core), please improve its description. Add labels to Figure 1 and refer to them in the text. Be sure that the labels are in the text (for instance, SC-DISCR, SC-main…). A more detailed scheme where the sensors and actuators are represented should be provided. 

Figure 4 is a general closed control system, it is not a “Basic Arduino microcontroller block diagram”. I think the figure should be removed. 

Regarding the code and functionality of the box, some flow charts should be provided with clear explanation through text labels. 

Is Figure 3 a starting procedure? You say that it represents the operating logic, but it seems a sort of initialization procedure. Again, a flow chart is better, with labels associated to different parts of the box. 

Section 3.2:  the first sentence should be moved to other place.

Caption of Figure 6: It is an interface, not a “software diagram”.

Minor: line 383, remove “center”

Generally speaking, the style of the whole paper should be reviewed, the number of sections reduced, many paragraphs should be removed to summarize the main contribution and focus on the design of the Box. A clearer description of the box is required. Provide details of the project as supplementary material.

Author Response

Please, see attached file.

Reviewer 2 Report

This is a relatively complete description of the building and operation of a behavioral box, with incorporation of both hardware and software. The resources and detail for other labs to be able to use (all specifications, details on building instructions, parts list, etc.) would be incredibly useful as an appendix. The authors state this will be disbursed on their website ("We have available all the necessary materials and 75
development documents and we will have it in a collaborative multi-user platform at our 76
Lab (www.natalnet.br), however, immediate use can be requested by e-mail to the authors." However I don't see why much of this can't be included in the appendix to this publication, to ensure this key/valuable information is in the public domain immediately on publication.

Author Response

Please, see attached file.

Round 2

Reviewer 1 Report

Thank you for addressing my comments. I think the work presented is valuable and likely useful for people whose research is in that area. I also doubt about the suitability if this paper as research article, but it could be published as other kind of paper instead, but this question must be decided by the editors of the journal.